# The Effect of Hormonal Therapy on the Behavioral Outcomes in 47,XXY (Klinefelter Syndrome) between 7 and 12 Years of Age

**DOI:** 10.3390/genes14071402

**Published:** 2023-07-06

**Authors:** Mary P. Hamzik, Andrea L. Gropman, Michaela R. Brooks, Sherida Powell, Teresa Sadeghin, Carole A. Samango-Sprouse

**Affiliations:** 1Department of Research, The Focus Foundation, 1251 W Central Ave. Suite A #190, Davidsonville, MD 21035, USA; 2Division of Neurogenetics and Developments Pediatrics, Children’s National Health System, 111 Michigan Ave. NW, Washington, DC 20010, USA; 3Department of Neurology, George Washington University, Washington, DC 20052, USA; 4Department of Economics, George Washington University, Washington, DC 20052, USA; 5Department of Human and Molecular Genetics, Florida International University, 11200 SW 8th St., Miami, FL 33199, USA; 6Department of Pediatrics, George Washington University, 2121 I St. NW, Washington, DC 20052, USA

**Keywords:** hormonal replacement therapy, early hormonal treatment, behavior, 47,XXY, sex chromosomal aneuploidy, SCA

## Abstract

47,XXY, also known as Klinefelter syndrome, is the most commonly occurring sex chromosomal aneuploidy (SCA). Hormonal replacement therapy (HRT) has been associated with improved neurodevelopmental capabilities in boys with 47,XXY, although studies investigating HRT’s possible positive effect on behavioral outcomes are scarce. This study explores the association between behavioral outcomes and HRT in boys ages 7–12. Patients were divided into 4 groups based on HRT status: untreated, early hormonal treatment (EHT), hormonal booster therapy (HBT), and both EHT and HBT. Analysis of Variance (ANOVA) and Kruskal–Wallis tests were conducted to determine group differences on the Child Behavior Checklist (CBCL) and the Behavior Rating Inventory of Executive Function (BRIEF). The treated groups were found to have better scores in emotional control, initiative, organization of materials, behavioral rating index, metacognition index, and global executive composite than the untreated group on the BRIEF. On the CBCL, the treated groups presented better scores for somatic complaints, social problems, thought problems, attention problems, aggressive behavior, internalizing problems, total problems, affective problems, somatic problems, ADHD problems, oppositional defiant problems, and sluggish problems in comparison to the untreated group. These results offer evidence that HRT, specifically the combination of both EHT and HBT, may be successful in mitigating some undesirable behavioral outcomes. Further research is necessary to determine the efficacy of the combination of EHT and HBT regarding dosage, specific ages, and long-term benefits.

## 1. Introduction

47,XXY, also known as Klinefelter syndrome, is the most commonly occurring sex chromosomal aneuploidy (SCA) variation with an incidence of approximately 1 in 500 to 660 live male births [1,2,3]. Medical features common to 47,XXY include increased height velocity, decreased fertility, truncal hypotonia, and androgen deficiency [1,4,5]. Although it varies by individual, the neurodevelopmental profile typically consists of developmental delay, language-based learning disabilities, neuromotor complications, developmental dyspraxia, early speech delays, and neuromotor dysfunction [4,6,7]. Boys with this chromosomal disorder may also be at increased risk for behavioral difficulties, including anxiety, depression, and executive dysfunction [8,9]. Behavioral challenges in boys with 47,XXY have been previously documented in the research literature. Tartaglia et al. (2010) [8] demonstrated that over half of their cohort met the criteria for “Withdrawn” on the Behavior Assessment System for Children (BASC), with 25% of the cohort also presenting increased scores for anxiety, depression, and somatization symptoms. Further, they found that 36% of their participants met the criteria for ADHD, with 34% showing primarily inattentive symptoms and 2% exhibiting a combined inattentive and hyperactive subtype [8]. Additionally, over 25% of this cohort showed elevated scores in all social responsiveness domains, except for social awareness, indicating potential concerns for social dysfunction in these boys [8]. Additionally, it was found that boys with 47,XXY displayed significantly more autistic traits and increased distress in social situations than control participants in this study [10]. 

Raznahan et al. (2014) [11] completed an MRI study that found that social cognition may be among the key areas of vulnerability in boys with 47,XXY. Using neuroimaging to view abnormalities in brain areas across various sex chromosome aneuploidies, including, 47,XXX, 47,XXY, 47,XYY, 48,XXYY, and 49,XXXXY, an algorithm revealed that the higher-order areas of the brain which are associated with social processing are impacted by sex chromosome aneuploidies, which may offer an explanation for the social dysfunction sometimes seen in individuals with extra chromosomes [11]. Other research studies have demonstrated deficits in inhibition, thought problems, aggression, and rule-breaking behavior in boys with 47,XXY. Deficits in inhibition can lead to difficulties regulating impulses, emotions, thoughts, and overall behavior [12]. 

It should be noted that these studies and various others reporting on the behavioral phenotype of 47,XXY are flawed due to several confounding variables. Behavioral outcomes in this population are often influenced by several factors, such as the size of the sample, timing of diagnosis, access to services, family history, and hormonal replacement therapy (HRT) [9,13,14]. An early study by Salbenblatt et al. (1981) [15] investigated a small cohort of boys with 47,XXY who were postnatally diagnosed, and found that they exhibited an increased incidence of poor relationships, anger issues, and adjustment difficulties in comparison to those prenatally diagnosed [15]. This study and several others offer evidence that early detection is favorable as it allows guardians to research and organize medical care providers and treatment options, and overall optimize a child’s developmental outcome.

Recently, research has focused on two forms of HRT with different timings of administration: early hormonal treatment (EHT) and hormonal booster therapy (HBT). EHT includes three injections, once per month, of 25 mg of testosterone enanthate between two and twenty-four weeks of age, while HBT consists of three injections, once monthly, of 50 mg of testosterone enanthate between five and eight years old [16,17]. Though not a direct focus of this study, hormonal replacement therapy has been associated with improving the overall neurodevelopment of boys with 47,XXY, including cognitive development, expressive and receptive language development, motor development, and working memory skills [16,17,18,19].

One study suggests that early hormonal treatment is effective at mitigating the language-based learning disabilities that are associated with 47,XXY [18]. This study explored auditory comprehension, expressive ability, and verbal intelligence in two cohorts of boys with 47,XXY: individuals who received early hormonal treatment, and individuals who had not. The two cohorts completed comprehensive age-appropriate expressive and receptive language assessments. The results indicated that there is a positive association between early hormonal treatment and improved language capabilities [18]. Another study explored neuromotor capabilities in boys with 47,XXY from birth to adulthood in relation to early hormonal treatment. The results show that the highest-treated combination group had the highest neuromotor function [19]. The positive impact of hormonal replacement therapy on various areas of development has led to the current exploration of its interaction with behavior in boys with 47,XXY.

In a study investigating anxiety symptoms, 40 boys had received hormonal replacement therapy prior to 10 years of age, and 40 had not. Boys treated with EHT and/or HBT showed significantly lower scores on the CBCL for social problems, thought problems, affective problems, total problems, anxiety/depression, ADHD problems, and OCD problems. In addition to these findings, individuals who were prenatally diagnosed had fewer indications of anxiety than individuals diagnosed postnatally [9]. In another study, boys who had received EHT and were evaluated at 36 and 72 months demonstrated significantly fewer behavioral difficulties and enhanced social behavioral abilities compared to those who had not been treated [14]. 

Boys with 47,XXY have a diverse presentation of neurodevelopmental skills and behavioral function. In previous studies, the behavioral profile of boys with 47,XXY has been associated with an increased incidence of anxiety disorders and attentional dysfunction [8,9,14]. Although the findings of recent research indicate that boys diagnosed with 47,XXY have behavioral dysfunction that may be associated with hormonal deficits, there is a paucity of literature regarding the interaction between HRT and behavior in young boys with 47,XXY. This study aims to further broaden the behavioral phenotype and investigate treatment options by exploring the effects of EHT and HBT on the behavioral phenotype in a large cohort of prenatally diagnosed boys with 47,XXY ages 7 to 12 years old.

## 2. Methods

### 2.1. Subjects

Eighty-five children between the ages of seven and twelve (*M* = 113.60 months, *SD* = 16.13 months) who were prenatally diagnosed with 47,XXY were referred for a comprehensive neurodevelopmental evaluation by their physicians, parents, and/or ancillary health care providers throughout the United States. To minimize ascertainment bias, The Focus Foundation (a non-profit research organization for X and Y chromosomal variations) funded evaluations for families when necessary. Medical records documenting a confirmed 47,XXY diagnosis and hormonal replacement therapy status were obtained prior to enrollment. Additionally, an in-depth intake was completed at the time of enrollment, which included demographic information, a three-generation family medical history, and pre-, peri-, and post-natal history on each child.

Each participant was evaluated by their community or tertiary care pediatric endocrinologist prior to enrollment in this study. Baseline testosterone levels were not drawn during the neurodevelopmental evaluation and are therefore not available for the majority of the subjects. Testosterone treatment was administered based on the pediatric endocrinologist’s clinical assessment of the subject and their endocrine needs in comparison to neurotypical boys.

The inclusion criteria for this study comprised a confirmed prenatal diagnosis of 47,XXY, and full-term pregnancy (i.e., >38 weeks). Participants with significant neonatal complications at the time of birth, including, but not limited to, neonatal seizures, significant birth complications requiring more than three days of hospitalization, or any brain abnormalities were excluded from this analysis. Participants who were found to have premature births (37 weeks or less), mosaicism, copy number variants (CNVs), post-natal diagnosis, and/or any other co-existing genetic disorders were also excluded from this study. The exclusion criteria were utilized to minimize confounding effects in order to determine the efficacy of HRT.

### 2.2. Behavioral Assessment

Behavioral testing was selected based on the subject’s chronological age. Each parent or guardian completed the Child Behavior Checklist (CBCL) and the Behavior Rating Inventory of Executive Function 5–18 years (BRIEF) to assess behavioral characteristics. On the CBCL, a *T*-score of 70 or above is considered in the clinical range at or above the 98th percentile. The borderline critical range lies between *T*-scores of 65 to 70, between the 95th and 98th percentile. *T*-scores 65 or below are considered normal at or below the 95th percentile. The BRIEF includes two index scores, the Behavioral Regulation Index (BRI) and the Metacognition Index (MI). The BRI includes “inhibit”, “shift”, and “emotional control”, while the MI includes “initiative”, “working memory”, “plan/organize”, “organization of materials”, and “monitor”. All subscale scores are combined for the Global Executive Composite Score. Higher *T*-scores suggest higher levels of dysfunction in each subtest, index, and the composite score.

### 2.3. Statistical Analyses

A protocol manager, who was blinded to the treatment statuses of the patients, scored each assessment while an offsite biostatistician, who had no interaction with the participants, performed the analyses. Subjects were segregated into four groups based on their testosterone treatment status: no-T, EHT, HBT, and EHT and HBT. Sixteen participants received no-T. Thirteen subjects were given EHT, which was prescribed by their pediatric endocrinologist, administered prior to two years of age, and included three injections of 25 mg of testosterone enanthate given once a month for three months. Twenty-eight subjects received HBT, which was prescribed by their pediatric endocrinologist, administered between five and eight years old, and consisted of three injections of 50 mg of testosterone, once per month for three months. Finally, twenty-eight participants received both EHT and HBT.

A mixed linear regression was completed as a mixed model accounting for both within-subject factors (different measurements at each visit) and between-subject factors. This is performed using statistics mixed command and controls for the lack of independence among repeated observations for a subject. Normality analyses including Bartlett’s Test of Equal Variances and Levene’s Test of Equal Variances revealed that the majority of variables were not normally distributed. Therefore, both an Analysis of Variance (ANOVA) and Kruskal–Wallis tests were completed to analyze group differences between the four treatment groups, as well as subsequent post-hoc tests. If multiple scores were available for an individual patient, all test scores falling within the appropriate age range for the subtest were used in the mixed linear regression analyses, while the first visit scores were utilized for the ANOVA and Kruskal–Wallis tests.

### 2.4. Editorial Policies and Ethical Considerations

Informed consent was obtained from each participant or parent of participants under the Western IRB-approved study protocol (20081226).

## 3. Results

The mixed linear regression revealed that the number of visits did not affect any dependent variable on either the BRIEF or CBCL. Additionally, there were no differences between groups for demographic variables (Table 1). Racial demographics for all participants are shown in Table 2. 

Seventy-eight total participants were analyzed for the BRIEF (*M* = 110.00 months, *SD* = 29.67 months): sixteen no-T (*M =* 110.00 months, *SD* = 29.67 months), fifteen EHT (*M =* 93.00 months, *SD* = 18.52 months), nineteen HBT (*M* = 113.74 months, *SD* = 26.52 months), and twenty-eight EHT and HBT (*M* = 115.00 months, *SD* = 20.75 months). An ANOVA test revealed a statistically significant difference for working memory on the BRIEF (*p* = 0.0784). A post hoc test found that scores between the EHT and EHT and HBT groups approached significance for working memory (*p* = 0.06), with the EHT and HBT group showing better scores.

On the Kruskal–Wallis test, group differences were found for emotional control, BRI, initiative, organization of materials, MI, and the global executive composite (Table 3). Specifically, the no-T group showed significantly depressed scores for emotional control (*p* = 0.004852), BRI (*p* = 0.0028), initiative (*p* = 0.0066), organization of materials (*p* = 0.0035), MI (*p* = 0.0042), and global executive composite (*p* = 0.0012).

Eighty subjects were assessed on the CBCL (*M* = 113.60 months, *SD* = 16.13 months): sixteen no-T (*M* = 116.69 months, *SD* = 15.92 months), thirteen EHT (*M* = 104.00 months, *SD* = 16.24 months), twenty-eight HBT (*M* = 115.82 months, *SD* = 16.35 months), and twenty-three EHT and HBT (*M* = 114.17 months, *SD* = 14.88 months). An ANOVA test found no significant differences between groups on the CBCL. In the Kruskal–Wallis test, group differences were revealed for somatic complaints, social problems, thought problems, attention problems, aggressive behavior, internalizing problems, total problems, affective problems, somatic problems, ADHD problems, oppositional defiant problems (ODD), sluggish problems, and post-traumatic stress problems (PTSD) (Table 4).

The no-T group showed significantly higher scores than the EHT and HBT group for somatic complaints (*p* = 0.00045), social problems (*p* = 0.00081), thought problems (*p* = 0.00013), attention problems (*p* = 0.0039), aggressive behavior (*p* = 0.0012), internalizing problems (*p* = 0.0042), total problems (*p* = 0.00016), affective problems (*p* = 0.00027), somatic problems (*p* = 0.00024), ADHD problems (*p* = 0.0078), ODD problems (*p* = 0.0076), sluggish problems (*p* = 0.0071), and PTSD problems (*p* = 0.00067).

Additionally, the no-T group presented with significantly higher scores in comparison to the EHT group in social problems (*p* = 0.0021), thought problems (*p* = 0.00040), total problems (*p* = 0.0052), sluggish problems (*p* = 0.0071), and PTSD problems (*p* = 0.0083). The EHT and HBT group showed decreased scores compared to the HBT group for attention problems (*p* = 0.0038), affective problems (*p* = 0.0047), and sluggish problems (*p* = 0.000081). The EHT group also showed better scores for somatic problems (*p* = 0.0038) in comparison to the HBT group.

## 4. Discussion

Boys with 47,XXY have an increased risk for anxiety, executive dysfunction, and other behavioral difficulties [8,9]. Previous studies have found that EHT has been associated with a positive influence on behavioral outcomes in 36- and 72-month-old children with 47,XXY [5,14]. This study supports and expands upon the finding that HRT may be a beneficial treatment in improving negative behavioral outcomes in boys between 7 and 12 years old with 47,XXY, particularly when both EHT and HBT are administered.

In the present study, externalizing problems were much less prominent than internalizing problems on the CBCL in boys with 47,XXY, regardless of treatment status. However, untreated boys in this study remained at a significantly higher risk for these internalizing behaviors. This is consistent with the findings of previous studies, which show that boys with 47,XXY have an increased risk for internalizing thoughts and anxiety disorders [8,9,12]. Participants who received both EHT and HBT demonstrated significantly lower risk for intrusive thoughts and anxiety, further supporting the potential association of benefits of hormonal treatment with improvement in behavior.

Previous literature has described the behavioral profile of young boys with 47,XXY by reporting an increased incidence of aggression, rule-breaking behavior, and criminality [8,19,20]. Select studies have previously attributed the cause of aggressive behavior to testosterone treatment in these boys [19]. However, our findings demonstrate that the group of boys who received both EHT and HBT present with significantly decreased scores for aggression and rule-breaking behavior on the CBCL. Further, the boys in our study show better stability in behavioral capabilities on both the BRIEF and CBCL than those who were untreated. These findings support the hypothesis that a combination of EHT and HBT may mitigate undesired externalizing outcomes and reduce maladaptive behaviors. Additionally, these findings support the possible link between the biological underpinnings of hormonal deficiency in 47,XXY, the behavioral phenotype, and improved outcomes with treatment.

The present study further supports hormonal treatment (specifically EHT and HBT) as a means of minimizing neurodevelopmental complications in boys with 47,XXY [5,9,21]. Although previous research has reported the benefits of EHT on motor and speech deficits in this population [5,22,23], few studies have investigated the effects of HBT on behavioral outcomes. Tran et al. (2019) [16] found the participants who received both EHT and HBT demonstrated significantly higher Working Memory Index scores on the WISC than their untreated counterparts. Working memory is a hallmark component of executive functioning that impacts the ability to self-monitor, and these deficits may exacerbate challenges in behavior [16]. Our findings, coupled with those of Tran et al. (2019) [16], suggest HBT may improve the underlying cognitive process of executive function and potentially reduce ADHD symptoms for boys with 47,XXY.

This retrospective cross-sectional study does have limitations. As with most rare genetic disorders or the SCAs rarely diagnosed, ascertainment bias makes it difficult to generalize our findings to the approximately 75% of individuals who remain undiagnosed. Further, because this study was not a randomized control trial, our results cannot be interpreted as directly causal. However, we have minimized significant confounding factors that commonly plague many studies on 47,XXY, including the timing of diagnosis, race, parental age, birth weight, and parental education. Additionally, although our sample of subjects consists of eighty-five children, dividing them into four groups to analyze the differences between hormone replacement therapy status does lead to a small cohort size. Future studies should expand upon our research in a larger cohort of boys with 47,XXY. Another limitation of this study is that since early hormonal treatment is administered during infancy, it is impossible to determine pretreatment differences in behavior because both the BRIEF and the CBCL do not assess behavior at this age. This study is the first of our knowledge to investigate the potential impact of EHT and HBT on the neurobehavioral profile in boys with 47,XXY. Our findings offer evidence for the efficacy of both EHT and HBT in mitigating internalizing and externalizing behaviors in this population. Additional research is warranted to determine how sex chromosome differences impact behavior in the brain, and equally important the impact of testosterone on behavioral function at various ages in boys with 47,XXY.

The use of cell-free DNA (cfDNA) of placental origin in the maternal plasma as a screening modality with the option to include the sex chromosomes has been made available to women beginning in 2011 [24]. These cfDNA tests are also known as noninvasive prenatal testing (NIPT) and evaluate the likelihood that the fetus has an autosomal trisomy for chromosomes 13, 18, or 21, with a low false positive rate and high sensitivity rate [25]. The cfDNA also allows for the detection of all children with SCA prenatally. Sensitivity rates are not yet as reliable as those for autosomal trisomy, but are rapidly improving.

Prenatal identification of a sex chromosome aneuploidy, specifically 47,XXY, has significant implications for children and families. Early detection and knowledge of clinical management among medical providers allow those affected to achieve optimal outcomes and appropriate neurodevelopmental progression and behavioral function. Therefore, this study further supports the need for early detection and possible hormonal treatment in children with 47,XXY. Further research is necessary to determine the optimal timing and dosage of HRT, as well as how this treatment may affect other neurodevelopmental aspects of 47,XXY.

## Figures and Tables

**Table 1 genes-14-01402-t001:** Demographics.

BRIEF (*N* = 78)									
	no-T		EHT		HBT		EHT and HBT		
N	16		15		19		28		
	*Mean*	*SD*	*Mean*	*SD*	*Mean*	*SD*	*Mean*	*SD*	*p*-value
CA	110.00	29.67	93.00	18.52	113.74	26.52	115.00	20.75	0.41
Birth wt.	6.00	0.89	7.11	0.78	5.86	1.07	7.14	1.21	0.31
Mat. age	39.29	3.15	37.25	5.39	38.50	3.39	34.63	4.87	0.50
Pat. age	41.60	3.13	35.86	5.61	41.40	4.72	34.71	3.15	0.28
CBCL (*N* = 80)									
	no-T		EHT		HBT		EHT and HBT		
N	16		13		28		23		
	*Mean*	*SD*	*Mean*	*SD*	*Mean*	*SD*	*Mean*	*SD*	*p*-value
CA	116.69	15.92	104.00	16.24	115.82	16.35	114.17	14.88	0.36
Birth wt.	6.50	1.08	7.00	0.77	6.07	0.98	6.94	1.30	0.34
Mat. age	37.64	4.03	37.00	7.01	37.28	3.42	34.95	4.26	0.41
Pat. age	38.60	4.25	37.89	7.41	40.14	5.51	35.20	3.05	0.32

Birth wt. (lbs.), CA (months), mat. age, and pat. age (years).

**Table 2 genes-14-01402-t002:** Racial demographics.

BRIEF and CBCL (*N* = 158)	
White	71.3%
Hispanic	7.5%
African American	3.70%
Other	10.0%

**Table 3 genes-14-01402-t003:** Kruskal–Wallis test BRIEF.

Kruskal–Wallis Test—BRIEF									
BRIEF	No-T		EHT		HBT		EHT and HBT		
	*Mean*	*SD*	*Mean*	*SD*	*Mean*	*SD*	*Mean*	*SD*	*p* Value
Inhibit	49.88	10.26	46.07	10.21	44.89	7.36	45.04	10.74	0.17
Shift	53.88	14.19	49.93	12.29	49.74	13.38	46.93	11.84	0.33
Emotional control	60.06	15.37	52.2	10.52	56.74	13.18	48.71	10.59	0.039 *
BRI	55.25	12.49	49	11.75	50.42	11.17	46.32	11.68	0.043 *
Initiative	35.81	8.52	46.87	11.1	50.89	8.83	47.21	13.1	0.046 *
Working memory	55.25	12.09	48.13	11.19	50.63	6.8	49.54	12.88	0.22
Plan/organize	55.31	11.66	46.62	14.49	51.11	9.09	47.43	14.1	0.045 *
Organization of materials	55.88	12.41	47.53	13.87	51.05	12.05	44.46	12	0.044 *
Monitor	52.88	12.12	44.67	14.03	45.42	15.36	42.57	14.15	0.071
MI	55.31	11.74	45.08	15	49.74	9.21	46	15.14	0.027 *
Global executive	55.94	12.14	46.38	14.32	50.21	10.31	45.89	14.32	0.012 *

* Significance at the *p* < 0.05.

**Table 4 genes-14-01402-t004:** Kruskal–Wallis test CBCL.

Kruskal–Wallis Test—CBCL									
CBCL	No-T		EHT		HBT		EHT and HBT		
	*Mean*	*SD*	*Mean*	*SD*	*Mean*	*SD*	*Mean*	*SD*	*p Value*
Activities	44.25	9.65	42.31	10.65	48	9.91	47.09	10.95	0.32
Social	46.07	9.38	50.25	5.03	48.31	8.03	52.87	8.23	0.14
School	41.31	9.69	47.64	9.72	42.57	6.98	44.2	9.7	0.2
Total competencies	43.38	10.29	47.89	10.23	46.62	10.17	50.5	10.98	0.21
Anxious/depressed	63.13	12.45	55.08	7.52	56.96	9.66	53.57	5.8	0.17
Withdrawn/depressed	58.88	8.13	53.38	6.08	56.86	6.22	54.13	6.3	0.054
Somatic complaint	61.44	10.15	53.92	6.95	55.29	7.22	52.57	4.99	0.0061 *
Social problems	60.94	9.37	53.31	4.77	57.36	8.88	53.09	4.5	0.0028 *
Thought problems	62.13	8.84	53.92	6.92	56.25	6.57	52.83	3.96	0.00033 *
Attention problems	56.16	7.68	59.44	9.24	57.29	7.86	53.83	3.96	0.0057 *
Rule-breaking behavior	55.25	7.9	51.69	2.81	52.29	3.79	51.74	2.96	0.46
Aggressive behavior	59.19	8.37	55.31	6.28	53.54	5.65	51.17	5.27	0.010 *
Internalizing problems	60.5	14.35	49.56	11.76	53.5	11.91	47.74	10.68	0.049 *
Externalizing problems	54.06	13.76	49.38	10.71	46.21	11.05	43.7	9.31	0.063
Total problems	60.25	12.24	48.38	11.71	52.64	10.26	45.87	9.72	0.0026 *
Anxiety problems	59.88	9.27	56.62	8.45	54.75	8.34	54.65	6.67	0.24
Affective problems	62.81	11.73	54.92	5.71	57.64	8.55	52.17	4.98	0.0027 *
Somatic problems	61.25	10.04	53.69	5.91	53.86	7.82	51.57	4.46	0.00065 *
ADHD problems	56.31	7.47	52.69	4.75	54.29	4.98	52.43	4.92	0.038 *
Oppositional defiant	59.56	8.1	56.62	6.47	54.12	5.22	53.21	4.3	0.046 *
Conduct problems	52.96	5.38	52.38	5.19	52.43	4.29	51.91	4.22	0.34
Sluggish problems	58.69	8.32	52.38	5.91	60.75	6.23	53.13	5.6	0.0001*
OCD problems	61.19	10.93	53.62	5.82	54.86	7.99	52.13	4.27	0.063 *
PTSD problems	62.31	11.09	53.92	7.72	55.86	8.69	52.57	5.64	0.0051 *

* Significance at the *p* < 0.05.

## Data Availability

No potentially identifiable human images or data are presented in this study.

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
