# Peer review of "The Effect of Hormonal Therapy on the Behavioral Outcomes in 47,XXY (Klinefelter Syndrome) between 7 and 12 Years of Age"

_genes, 2023, doi:10.3390/genes14071402_

Round 1

Reviewer 1 Report

The current report, entitled, “The effect of hormonal therapy on the behavioral outcomes in 47, XXY” claims that, “These results offer evidence that HBT….” May be successful in mitigating some undesirable behavioral outcomes.”  The report includes findings from 3 groups of males with XXY treated with hormone replacement (early, booster or combination) compared to an untreated group, comparing results on 2 parent questionnaires reporting on aspects of behavioral functioning, in order to make these conclusions.

While the study reports on these behavioral questionnaires, and indicates the limitation that it is a retrospective cross-sectional study, it is unclear how the study was conducted.   How was the sample selected?  When were the subject enrolled? Which pediatric endocrinologists were contacted, how were they chosen and how representative are they?   Of the many possible confounding factors that may account for differences between groups, what could else could be different between treated groups and the untreated group?  The study claims to account for socioeconomic status, but information on that, or race/ethnicity of the groups was not found. It is also unclear why the authors refer to the behavioral assessments as “comprehensive” on line 108 when it only consisted of 2 parent questionnaires. A comprehensive behavioral assessment would need to include assessment of developmental level as well, since factors such as cognitive impairment and language level may impact results on the behavioral questionnaires used.

Besides these lacking important details, the most concerning aspect of this report was the overreaching in claims made, based on wording chosen.  In no way did this report include a treatment study, and importantly, parents were not blind to whether their children received treatment, so making claims like “offer evidence for the efficacy of both EH and HBT” are not appropriate (line 236). 

Regarding statistics conducted, it is unclear why both ANOVAs and Kruskal-Wallis tests were run and reported for all variables – this does not make sense as written.  Non-parametrics should be used when residuals are not normally distributed, but in this case, with truncated distributions of scores on these measures, ANOVAs are definitely not appropriate. How was chronological age on the BRIEF controlled for in the Kruskal-Wallis analyses?  Table 1 seems to show the BRIEF twice, (not sure what is at the bottom), but the significant difference in CA for the BRIEF is not indicated by the p-value shown.

A very concerning aspect of the way the results/interpretation was discussed is the fact that while there may have been some differences between groups on scales of these measures, how can the authors claim “beneficial treatment in improving negative behavioral outcomes” (line 194), when a) these children are still quite young, but more importantly, b) they were reporting on mean scores, rather than percent in the clinically significant range?  In other words, reporting on lower scores for a particular group, when all groups are in the “normal range” is not particularly meaningful.  If the measure doesn’t indicate these scores are of clinical concern, then why does it matter if they are “statistically significantly lower”?  While other reports along these lines comparing hormone treated patients to those without have not explored pretreatment differences, it really is difficult to make sense of these types of findings at this point without being able to control at least in some way for differences that may have existed between those who are treated vs. those who are not.

Further, the discussion of internalizing vs. externalizing, including “externalizing behaviors were much less prominent than internalizing behaviors on the CBCL” similarly did not make sense, since neither of these was in the clinically significant range.

n/a

Reviewer 2 Report

Thank you for this very relevant and interesting paper. It is well written and I have only remarks about your tables:

table1: there is a part missing regadring the information about BRIEF

table 2&3: they are rather difficult to read, maybe you can find a more visual method to present them
